# Multimodal Treatment of Malignant Pleural Mesothelioma: Real-World Experience with 112 Patients

**DOI:** 10.3390/cancers14092245

**Published:** 2022-04-30

**Authors:** Arnulf Holzknecht, Oliver Illini, Maximilian J. Hochmair, Dagmar Krenbek, Ulrike Setinek, Florian Huemer, Erwin Bitterlich, Christoph Kaindl, Vladyslav Getman, Ahmet Akan, Michael Weber, Gunther Leobacher, Arschang Valipour, Michael R. Mueller, Stefan B. Watzka

**Affiliations:** 1Karl Landsteiner Institute for Thoracic Oncology, Klinik Floridsdorf, 1210 Vienna, Austria; arnulfd@googlemail.com (A.H.); vladyslav.getman@gesundheitsverbund.at (V.G.); dr.ahmet1977@gmail.com (A.A.); michael.mueller@gesundheitsverbund.at (M.R.M.); 2Division of Thoracic Surgery, Karl Landsteiner Institute of Thoracic Oncology, Klinik Floridsdorf, 1210 Vienna, Austria; 3Department of Respiratory and Critical Care Medicine, Klinik Floridsdorf, 1210 Vienna, Austria; oliver.illini@gesundheitsverbund.at (O.I.); maximilian.hochmair@gesundheitsverbund.at (M.J.H.); arschang.valipour@gesundheitsverbund.at (A.V.); 4Karl Landsteiner Institute for Lung Research and Pulmonary Oncology, Klinik Floridsdorf, 1210 Vienna, Austria; 5Institute for Pathology, Klinik Floridsdorf, 1210 Vienna, Austria; dagmar.krenbek@gesundheitsverbund.at; 6Institute for Pathology and Microbiology, Klinik Ottakring, 1160 Vienna, Austria; ulrike.setinek@gesundheitsverbund.at; 7Division of Pulmonology, Klinik Penzing, 1140 Vienna, Austria; florian.huemer@gesundheitsverbund.at; 8Division of Pulmonology, Salzkammergutklinikum Voecklabruck, 4840 Voecklabruck, Austria; erwin.bitterlich@ooeg.at; 9Division of Surgery, Salzkammergutklinikum Voecklabruck, 4840 Voecklabruck, Austria; christoph.kaindl@ooeg.at; 10Department of Medical Imaging and Image-Guided Therapy, Medical University of Vienna, 1090 Vienna, Austria; michael.weber@meduniwien.ac.at; 11Institute of Mathematics and Scientific Computing, University of Graz, 8010 Graz, Austria; gunther.leobacher@uni-graz.at; 12Division of Thoracic Surgery, Sigmund Freud University, 1020 Vienna, Austria; 13Division of Surgery, Paracelsus Medical University, 5020 Salzburg, Austria

**Keywords:** malignant pleural mesothelioma, chemotherapy, surgery, radiotherapy, survival

## Abstract

**Simple Summary:**

To evaluate the best possible treatment of malignant pleural mesothelioma—a cancer whose development is associated with asbestos exposure—an analysis of 112 consecutive patients treated at a high-volume center in Vienna (Austria) was conducted. The average survival of all patients was 16.9 months after diagnosis. Of the patients who underwent combined chemotherapy and lung-preserving surgery, 29% were still alive 5 years after diagnosis. In statistical analysis, combined chemotherapy and surgery, epithelioid tumor subtype, early tumor stage and the absence of relevant comorbidities were found to be favorable factors for survival. Therefore, the best possible treatment for malignant pleural mesothelioma should incorporate multiple therapeutic approaches.

**Abstract:**

Malignant pleural mesothelioma (MPM) is a rare pleural cancer associated with asbestos exposure. According to current evidence, the combination of chemotherapy, surgery and radiotherapy improves patients’ survival. However, the optimal sequence and weighting of the respective treatment modalities is unclear. In anticipation of the upcoming results of the MARS-2 trial, we sought to determine the relative impact of the respective treatment modalities on complications and overall survival in our own consecutive institutional series of 112 patients. Fifty-seven patients (51%) underwent multimodality therapy with curative intent, while 55 patients (49%) were treated with palliative intent. The median overall survival (OS) of the entire cohort was 16.9 months (95% CI: 13.4–20.4) after diagnosis; 5-year survival was 29% for patients who underwent lung-preserving surgery. In univariate analysis, surgical treatment (*p* < 0.001), multimodality therapy (*p* < 0.001), epithelioid subtype (*p* < 0.001), early tumor stage (*p* = 0.02) and the absence of arterial hypertension (*p* = 0.034) were found to be prognostic factors for OS. In multivariate analysis, epithelioid subtype was associated with a survival benefit, whereas the occurrence of complications was associated with worse OS. Multimodality therapy including surgery significantly prolonged the OS of MPM patients compared with multimodal therapy without surgery.

## 1. Introduction

Malignant pleural mesothelioma (MPM) is a malignant pleural disease whose development is associated with antecedent asbestos exposure [1,2]. The incidence of MPM varies between 1 and 30 cases/million/year, with an increasing tendency [1]. According to the newest estimates, 38,400 deaths worldwide can be ascribed to MPM [3], in which 95% of those affected had been exposed to asbestos [4]. The latency time after asbestos exposure is approximately 15–45 years [5,6]. Because of this, two-thirds of all MPM patients are aged 50–70 years at the time of the first diagnosis [7]. Patients who do not receive treatment have a survival time of 4–12 months after the first diagnosis [8]. Moreover, a high percentage of MPM patients are diagnosed in an advanced tumor stage [9], which is not only associated with worse survival [10] but also makes the application of radical therapeutic approaches unreasonable.

Different prognostic factors such as histologic subtype, gender, tumor stage, age, performance status, weight loss, anemia, leukocytosis, thrombocytosis or elevated serum lactate dehydrogenase have been reported [1]. Different treatment modalities such as surgical modalities for macroscopic cytoreduction, including extended pleurectomy/decortication (EPD), pleurectomy/decortication (P/D) [11] (referred to as lung-preserving surgery (LPS) or extrapleural pneumonectomy (EPP)) [12], chemotherapy [1,13] and radiotherapy [14], have been adopted in order to improve patients’ long-term survival [1]. The best results can be found for patients receiving multimodality therapy, usually consisting of neoadjuvant chemotherapy, followed by surgery and adjuvant radiotherapy [11,12,13,14]. Through combining these modalities, a median survival of 17–38 months for patients in tumor Stage I to 7–24 months in tumor Stage IV can be achieved [10,15,16,17,18,19,20,21].

However, since prospective randomized trials are difficult to perform due to the heterogeneity of the disease, there is still no solid evidence regarding the optimal combination of treatment modalities within the multimodal approach (neoadjuvant or adjuvant chemotherapy, lung-preserving or -destroying surgery, neoadjuvant or adjuvant radiotherapy) [22,23] or regarding which patients are most likely to benefit from which treatment [1]. The ongoing multicentric, prospective and randomized MARS-2 trial attempts to answer these questions by comparing the overall survival of MPM patients after (extended) pleurectomy/decortication and chemotherapy with chemotherapy alone [24]. In anticipation of the upcoming results of the MARS-2 trial, in the present study, we attempted to answer the question whether a favorable survival outcome of MPM could be achieved by a standardized institutional treatment algorithm, based on an analysis of 112 consecutive patients treated at our referral center.

## 2. Materials and Methods

### 2.1. Study Design

The medical records of all consecutive MPM patients treated at a referral center for thoracic oncology and thoracic surgery in Vienna (Austria) from January 2000 to May 2020 were analyzed. Patients with any other forms of malignant or benign mesothelial disease were excluded. The charts were searched for the patients’ age, comorbidities, affected thoracic side, histologic subtype, tumor stage, lymph node metastases, treatment sequence, type of operation, surgical radicality, postoperative complications and survival. The study was approved by the institutional review board of the city of Vienna (EK_14_030_VK). According to Austrian laws, informed consent for each patient was not necessary for this retrospective analysis. The study was conducted according to the principles of the Declaration of Helsinki.

### 2.2. Diagnostic Approach and Treatment Characteristics

If MPM was suspected after clinical evaluation and examining the medical history, a thoracic CT-scan was performed; subsequently, patients underwent biopsy via diagnostic video-assisted thoracoscopic surgery (VATS) or thoracotomy. Patients suffering from recurrent pleural effusion were treated with talcum pleurodesis at the time of VATS. With a histological diagnosis of MPM, patients received a PET scan as part of the staging and, occasionally, diagnostic mediastinoscopy if the PET scan indicated the presence of mediastinal lymphadenopathy. For staging, the TNM system (version 2010) was used [25].

Standardized neoadjuvant treatment recommendations with curative intent included four cycles of cisplatin or carboplatin/pemetrexed (or platin/gemcitabine, prior to the approval of pemetrexed [13]) or up to 6 cycles in a palliative setting. After neoadjuvant chemotherapy, patients showing good treatment response received surgery, by either pleurectomy/decortication (P/D), extended pleurectomy/decortication (EPD) or extrapleural pneumonectomy (EPP), depending on each patient’s clinical presentation and disease characteristics. Wherever possible, the surgical approach aimed to preserve the lung, thus an EPD or a P/D was performed. Extrapleural pneumonectomy (EPP) was only performed when not only the pleura but also the lung parenchyma was affected, and thus performing LPS would not have removed the entire tumor mass. If the spatial distribution of the disease allowed exclusive pleural and mediastinal radiotherapy without excessive damage to the preserved lung parenchyma, patients received adjuvant radiotherapy at a specialized academic referral center. Adjuvant radiotherapy was performed as intensity-modulated radiation therapy (IMRT) with a dose of 54–60 Gy (2 Gy per daily fraction). Patients not responding to neoadjuvant chemotherapy or patients who were unsuitable for surgery underwent subsequent radiotherapy if feasible, plus follow-up or the best supportive care only (Figure 1). If MPM was not suspected at the time of surgery, patients received adjuvant chemotherapy following the pathological diagnosis instead of neoadjuvant chemotherapy.

Except for a few modifications, EPP and EPD were essentially performed as described elsewhere [26,27]. Briefly, after performing a posterolateral thoracotomy, the parietal pleura was mobilized from chest wall, mediastinum and diaphragm, possibly without opening the pleural sac. In the case of EPP, the pericardium was opened, and both pulmonary veins, the pulmonary artery and the main bronchus were exposed and subsequently divided; afterwards, the remainder of the pericardium as well as the diaphragm were excised en bloc and removed together with the pleural sac encompassing the whole lung. In the case of EPD and P/D, however, the parietal pleural sac together with the visceral pleura were resected completely from the lung (including the interlobar fissures), as well as from the pericardium and diaphragm. In case of disease involvement in the pericardium and diaphragm, both structures were resected entirely and en bloc, in which case, the procedure was termed EPD; otherwise, it was termed P/D. In order to avoid spilling tumor cells, in all cases of EPD and P/D, a non-incisional procedure, as described recently [28], with preservation of the integrity of the parietal–visceral pleural sac was aimed at. After the resection, all accessible mediastinal lymph nodes were removed, and reconstruction of the diaphragm and pericardium commenced: in the place of the diaphragm, a suitably fashioned GORE DUALMESH patch was implanted, and in the place of the pericardium, an absorbable VICRYL mesh was used. Hemostasis was achieved by irrigation with a hyperthermic isotonic saline solution (42 °C) and/or the adoption of irrigated bipolar sealing (Aquamantys, Medtronic, Tolochenaz, Switzerland). To treat the pleurectomized lung surface, autologous fibrin (Vivostat, Alleroed, Denmark) was sprayed on the area. After routine insertion of three large-bore chest tubes and closure of the chest, the patient was transferred to the ICU.

### 2.3. Statistical Analysis

Overall survival (OS) was defined as the time from initial diagnosis to date of death or censoring date. The date of censoring was the date of the last follow-up.

Median OS was calculated using the Kaplan–Meier estimator and a confidence interval (CI) of 95%. Median follow-up was calculated using the reversed Kaplan–Meier estimator. The independence of the patient subgroups was tested using Fisher’s exact test. Univariate analysis using Cox regression with a level of significance of 5% (chi square *p* = 0.05) was conducted for factors of potential prognostic relevance, including age, sex, tumor stage, histologic subtype, treatment and the presence of comorbidities, as recommended by the current guidelines of the European Society of Thoracic Surgeons (ESTS) [1]. Factors that showed a significant impact on survival in the univariate analysis were included in the multivariate analysis, which was performed by Cox regression using a stepwise algorithm. All statistical analysis were conducted using SPSS (IBM, Armonk, NY, USA, version 25).

## 3. Results

Between 2000 and 2020, 112 patients with MPM were treated. Eighty-eight patients were male (78.6%); the median age at the time of the diagnosis was 67.5 years. Most patients (77.7%) had an epithelioid subtype and were diagnosed with Stage IV disease (39.3%). Overall, 40 patients (35.7%) had LPS with curative intent, encompassing either EPD (*n* = 34; 30.4%) or P/D (*n* = 6; 5.4%). Nine patients (8.0%) were treated by EPP. Sixty-nine patients (61.6%) had comorbidities documented in their medical history (Table 1).

In total, 85 patients (75.9%) received neoadjuvant chemotherapy. Of these patients, 49 (43.8%) had a tumor resection with curative intent, of which 10 (8.9%) received subsequent adjuvant radiotherapy, and thus completed classic trimodal therapy. Thirty-six patients (32.1%) did not receive surgery after neoadjuvant chemotherapy. Subsequently, 10 (8.9%) of these patients were treated by radiotherapy. In eight patients (7.1%), the diagnosis of MPM was not expected on the basis of clinical and radiological assessments preoperatively. These patients underwent surgical treatment followed by adjuvant chemotherapy; thus, all patients who underwent surgery received chemotherapy (adjuvant or neoadjuvant) (Figure 2).

The types of comorbidities are shown in Table 2. Notably, patients who subsequently underwent curative surgery suffered significantly more often from arterial hypertension than patients who were treated with palliative intent.

In total, 29 of all 49 surgically treated patients (59.2%) developed postoperative complications (Table 3). The most common complications after LPS were blood loss requiring the application of red cell concentrate (RCC) (*n* = 10; 25.0%) or postoperative air leaks (*n* = 8; 20.0%). There were no deaths reported in the first 30 days after the operation.

The median follow-up time was 83.1 months (95% CI: 47.2–118.9). The median and stage-independent OS of the whole cohort was 16.9 months (95% CI: 13.4–20.4). One-year OS was 64.1% (*n* = 68), 3-year OS was 19.8% (*n* = 19) and 5-year OS was 13.1% (*n* = 10) (Table 4).

Patients with the epithelioid subtype had a median OS of 20.3 months (95% CI: 16.9–23.7), which was significantly longer (*p* < 0.001) than the survival of patients with the biphasic or non-epithelioid subtype, with a median OS of 6.8 months (95% CI: 3.2–10.4). Patients receiving surgery and additional therapy (adjuvant or neoadjuvant chemotherapy and/or radiotherapy) showed a significantly (*p* < 0.001) better outcome, with a median OS of 22.7 months (95% CI: 17.2–28.2), compared with patients who had not received additional oncological treatment (median OS: 9.9 months, 95% CI: 6.7–13.1). This difference was also significant for patients receiving chemotherapy followed by surgery without radiotherapy, with an OS of 21.2 months (95% CI: 14.2–28.2; *p* = 0.002). Patients who completed trimodal therapy consisting of neoadjuvant chemotherapy, surgery and adjuvant radiotherapy had a median OS of 30.8 months (95% CI: 7.2–54.4).

The median OS in patients treated with LPS was 25.5 months (95% CI: 15.1–35.9), while it was 12.3 months after EPP (95% CI: 12.1–13.5). Patients’ 3 yr OS after LPS was 38.3%, with a 5 yr OS of 29.2%, while EPP had a 3 yr OS of 0%. This difference in survival between P/D and EPP patients was significant (*p* = 0.017). All surgically treated patients had a 3 yr OS of 31.0% (*n* = 14) and a 5 yr OS of 23.6% (*n* = 9) (Figure 3).

The median OS in tumor Stage I was 42.3 months (95% CI: 17.8–66.8), while it was 18.1 months in Stage II (95% CI: 16.0–20.2), 17.4 months in Stage III (95% CI: 15.1–19.7) and 8.5 months (95% CI: 5.0–9.0) in Stage IV; these differences were statistically significant (*p* < 0.001) (Figure 4).

In the univariate analysis, any form of surgical intervention with curative intent was significantly linked to better survival (*p* < 0.001), while postoperative complications were linked to worse survival (*p* = 0.02). All patients receiving multimodal therapy showed a significantly better survival outcome (*p* < 0.001) than patients receiving other forms of treatment. There were also significantly higher survival rates for the epithelioid subtype (*p* < 0.001), the early tumor stage (*p* = 0.002), lung-preserving surgery (*p* < 0.001), and female sex in comparison to male sex (*p* = 0.021) and age younger than 55 years (*p* = 0.007) (Table 5).

In surgically treated patients, epithelioid subtype (*p* = 0.008), the application of LPS (*p* = 0.017), and female sex (*p* = 0.032) were linked to better survival (Table 6).

In multivariate analysis, only the epithelioid subtype and early tumor stage had a significantly positive effect, while cardiac decompensation and arterial hypertension had a significantly negative impact on OS (Table 7).

## 4. Discussion

Malignant pleural mesothelioma is a rare malignancy which is characterized by poor prognosis, and the optimal treatment strategy is still under debate [29,30,31]. In this retrospective analysis of 112 consecutive patients treated in a single center, several prognostic factors which significantly affected OS have been identified. The results indicate that the best treatment results can be achieved by multimodal therapy consisting of adjuvant chemotherapy, surgery and radiotherapy, where applicable. With a median OS of 30.8 months for trimodality therapy, 21.4 months without radiotherapy and 23.3 months for patients receiving any form of multimodality treatment (consisting of surgical intervention and radiotherapy or chemotherapy or both), all patients who received surgery with curative intent and additional treatment achieved an impressive survival outcome. These findings highlight the necessity of a standardized institutional treatment algorithm which not only standardizes patient treatment but also facilitates interinstitutional comparisons of the results. However, the results of the MARS-2 trial [24] will reveal whether improved chemotherapy protocols can efficiently compete with previously established multimodal therapy.

Due to the low incidence of MPM, studies with large representative patient populations are rare. A possible comparison for our results can be found in the International Association for the Study of Lung Cancer (IASLC) Mesothelioma Database, which contains 3101 patients from 15 centers [17]. A median OS of 20 months for patients treated with multimodality therapy is reported, which is comparable with our results for these subgroups. For stage-dependent median OS, we found 42, 18, 17 and 9 months for tumor Stages I–IV compared with 21, 19, 16 and 12 months reported by the IASLC, respectively, assuming a remarkable survival, especially for our early-stage patients. Interestingly, our patients were slightly older (a median age of 67.5 compared with 63 years) and fewer patients received surgery with curative intent (42.6% compared with 64.5%). The percentage of female patients and the percentage of male patients (78%) treated in our center was almost equal to the 79% male patients published by the IASLC. However, it should be noted that in the cohort published by the IASLC, 207 of the 1162 patients in the curative intent group received only surgery without chemotherapy or radiotherapy, while all our surgically treated patients received either adjuvant or neoadjuvant chemotherapy.

In another recently published analysis of 560 patients from the Spanish Malignant Pleural Mesothelioma Database, a median OS in the whole cohort of 13.0 months was presented, while we report a median OS of 16.9 months for unselected patients [32]. This difference can partly be explained by the lower surgery rate with curative intent (29%) and low rates of tri- and bimodal regimens (3% and 11% compared with 9% and 44%). Moreover, our cohort contained a higher percentage of patients with the epithelioid subtype (89% compared with 62% in both), a fact that may also have contributed to the better survival in some subgroups [17,32]. The observation that our patients who were in earlier tumor stages showed noticeably good outcomes can be made in comparison with most other reported studies as well, as the literature reports results ranging from 17–21 months for Stage I, 11–33 months for Stage II, 11–31 months for Stage III and 8–24 months for Stage IV [10,15,16,17,18,19,20,21].

The prognostic factors we found in the multivariate analysis indicate that the patients who were most likely to benefit from MPM therapy were those with epithelioid subtype, receiving EPD or P/D, and who did not have arterial hypertension and in which cardiac decompensation was avoided. The univariate analysis also suggested that female patients, younger patients and patients receiving lung-preserving surgery, especially as part of multimodal therapy, are more likely to benefit. Given these data, it remains unclear if the better median survival is because of the higher percentage of epithelioid subtypes, other unknown underlying patient features or factors specific to our center, such as the surgeons’ experience and equipment, or a more efficient treatment algorithm (see Table 8, which gives an overview of survival in other published cohorts).

Studies such as the meta-analysis performed by Taioli et al. [33], the retrospective study by Flores et al. [16] and the Mesothelioma and Radical Surgery (MARS) trial by Treasure et al. [34] suggested that EPP is more harmful to the patients than the lung-preserving procedures (EPD or P/D), leaving the question open as to which surgical intervention is to be preferred in which case [16]. Other studies have reported a similar outcome between P/D and EPP [30,35]. However, the better survival seen in our patients treated with trimodality therapy, or chemotherapy combined with surgery (either by EPD or P/D), as well as in patients treated with any form of surgical intervention seems reasonable, as macroscopic complete resection was previously defined as an important goal in MPM treatment [36]. However, in the univariate analysis, patients receiving EPP showed worse survival than patients receiving EPD or P/D. It must be noted that only nine patients in our cohort were treated with EPP, which, in accordance with our institutional policy, was only adopted when EPD or P/D were not accomplishable due to massive disease involvement of the lung parenchyma. In addition, patients treated with EPP had higher rates of advanced tumor stages than those treated with LPS (Stage III/IV: 75% compared with 52%). Because of these obvious selection biases, the real value of EPP in the treatment of MPM cannot be commented on reliably on the basis of our data. The finding that multimodal therapy including surgery significantly prolonged the OS in the univariate analysis must be interpreted with caution, as the multivariate analysis that adjusted for confounders did not identify the addition of surgery as an independent factor influencing the OS, and the epithelioid subtype was more frequent in the surgery subgroup. It also remains unclear how patients in which macroscopic complete resection cannot be performed (e.g., patients in advanced tumor stages) can be treated best. Furthermore, it is uncertain whether the survival benefit of patients receiving multimodal therapy is attributable to a selection bias, since surgical therapy, especially in combination with other treatment modalities, is reserved for the fittest patients in earlier tumor stages. On the other hand, as we had adopted a standardized institutional algorithm to which the patients had to adhere, the principal selection bias in our center was certainly less significant than in other centers that make case-by-case decisions.

In our study, information about the systemic treatment regimens in the case of recurrence of disease has not been analyzed. Recent advances including immunotherapy as part of MPM treatment highlight the question if and how immunotherapy can be implemented into a multimodal regimen and improve the outcome [37,38]. Another limitation of our analysis is the lack of pathological grading data: it is now well recognized that these are able to predict survival [39]. However, as the grading procedure is relatively time-consuming, it was not implemented into the routine pathological work-up. This might change in the future, when ever-larger panels of histological and molecular biological markers become analyzable by semi-automatic or even fully automatic methods. The retrospective study design is also an obvious limitation; however, this is shared with the majority of studies published on this subject. Finally, our complication report system is not universally accepted. As a matter of fact, it seems that our patients developed more complications compared with other studies, as 56.8% of our patients treated by EPD or P/D and 88.9% of all patients treated by EPP suffered from postoperative complications, while the literature reports lower complication rates ranging from 15.6 to 43% for P/D or EPD [40,41] and from 50.6 to 60% for EPP [40,42]. However, as there is yet no consensus on which complications should be reported, it is unclear if the higher rates in our patients are due to an actual higher occurrence of complications or a wider range of reported complications in our cohort. For better comparison of studies, in future guidelines, it should be specified which complications should be documented, so that rarer complications are not neglected because of the low incidence of MPM. In that way, it could be evaluated if special precautions against certain complications significantly improve patients’ survival.

## 5. Conclusions

In summary, it is of paramount importance to optimize multimodal MPM therapy to improve patients’ outcomes. At present, the optimal sequence of modalities is still unclear, and solid evidence is still not available. However, adopting a standardized institutional algorithm and strictly adhering to it might pave the way for the definition of universally accepted therapeutic standards.

## Figures and Tables

**Figure 1 cancers-14-02245-f001:**
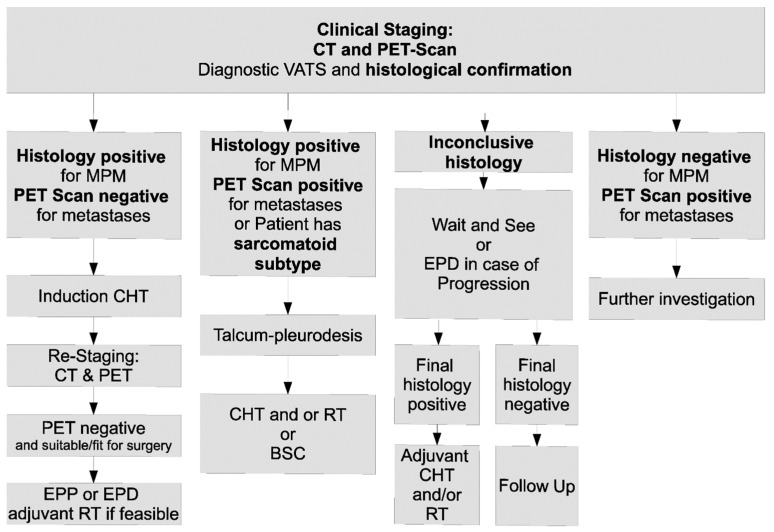
Algorithm of the treatment approach used in our institution. BSC best supportive care, CHT chemotherapy, CT computed tomography, EPD extended pleurectomy/decortication, EPP extrapleural pneumonectomy, MPM malignant pleural mesothelioma, PET positron emission tomography Scan, RT radiotherapy.

**Figure 2 cancers-14-02245-f002:**
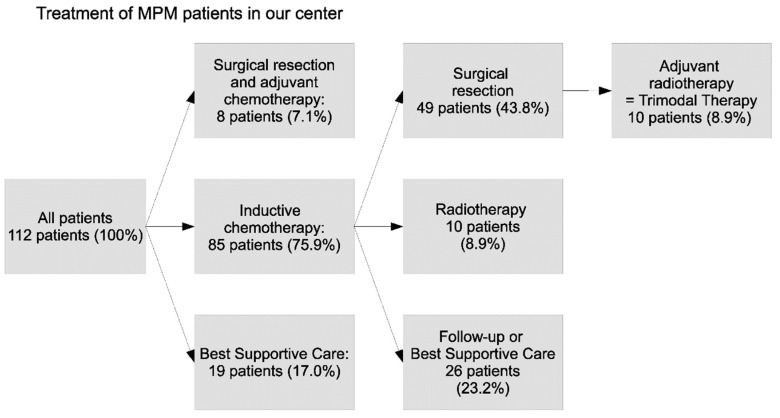
Treatment of MPM patients in our institution. In total, 112 patients were analyzed. Best supportive care also included all patients who did not receive chemo- or radiotherapy after a diagnostic VATS or debulking. MPM malignant pleural mesothelioma, VATS video-assisted thoracoscopic surgery.

**Figure 3 cancers-14-02245-f003:**
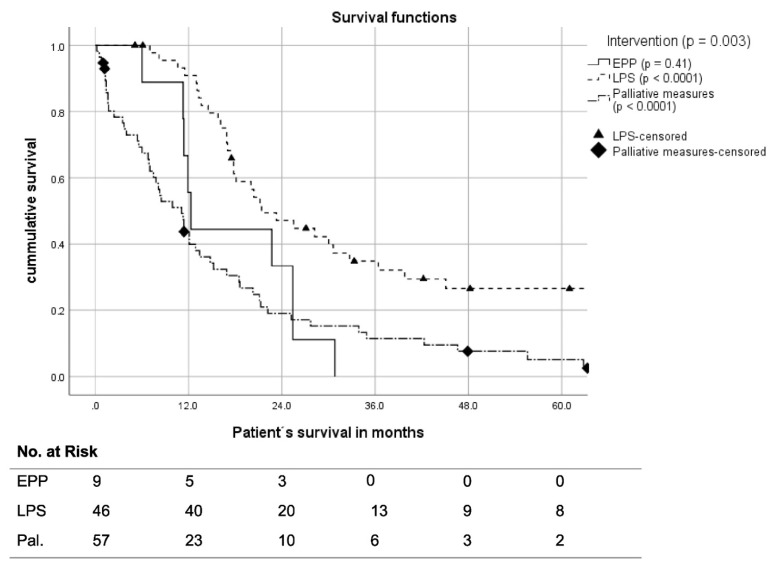
Kaplan–Meier curves of patient survival according to the therapy regimen received. EPP extrapleural pneumonectomy, LPS lung-preserving surgery, Pal. palliative measures.

**Figure 4 cancers-14-02245-f004:**
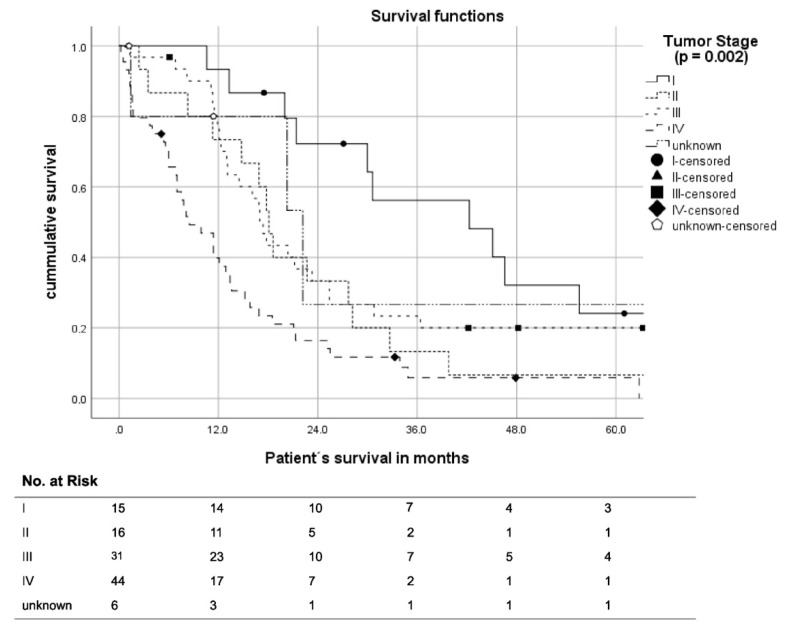
Kaplan–Meier curves of patient survival according to tumor stage.

**Table 1 cancers-14-02245-t001:** Patients’ characteristics. The subgroups of patients were tested for independence; the *p*-value was determined by Fisher’s exact test. EPD extended pleurectomy/decortication, EPP extrapleural pneumonectomy, LPS lung-preserving surgery, VATS video-assisted thoracoscopic surgery, SD standard deviation, statistically significant *p*-values were printed in bold.

Patient Characteristics N (%)	Total Cohort *n* = 112	Conservative Therapy *n* = 63 (56.3%)	Surgical Therapy with Curative Intent *n* = 49 (43.8%)	*p*-Value Fisher’s Exact Test
Age at diagnosis (per 1 year increase) Mean age (SD) Range	65.5 (11.0) 37–86	68.8 (10.2) 37–86	61.2 (10.5) 37–80	0.10
Sex Female Male	24 (21.4%) 88 (78.6%)	10 (15.9%) 53 (84.1%)	14 (28.6%) 35 (71.4%)	0.36
Side Left Right	68 (60.7%) 44 (39.3%)	37 (58.7%) 26 (41.3%)	31 (63.3%) 18 (36.7%)	0.34
Diagnosis by VATS Other	103 (92.0%) 9 (8.0%)	60 (95.2%) 3 (4.8%)	43 (87.8%) 6 (12.2%)	0.20
Histological subtype Epitheloid Biphasic Sarcomatoid Lymphohistiocytoid mesothelioma Not determined	87 (77.7%) 21 (18.8%) 2 (1.8%) 1 (0.9%) 1 (0.9%)	43 (68.3%) 16 (25.4%) 2 (3.2%) 1 (1.6%) 1 (1.6%)	44 (89.8%) 5 (10.2%) 0 0 0	**0.004**
Surgical treatment EPP LPS Any surgery with palliative intent		8 (7.1%)	9 (8.0%) 40 (35.7%)	**<0.001**
Pathological stage Stage I Stage II Stage III Stage IV Unknown	15 (13.4%) 16 (14.3%) 31 (27.7%) 44 (39.3%) 6 (5.4%)	3 (4.8%) 8 (12.7%) 9 (14.3%) 38 (60.3%) 5 (7.9%)	12 (24.5%) 8 (16.3%) 22 (44.9%) 6 (12.2%) 1 (2.0%)	**<0.001**
Comorbidities	69 (61.6%)	40 (63.5%)	29 (59.2%)	0.85
Complications	46 (41.1%)	17 (27.0%)	29 (59.2%)	0.002

**Table 2 cancers-14-02245-t002:** Comorbidities in the whole cohort. The *p*-values and hazard ratios were determined by univariate Cox regression. NIDDM non-insulin-dependent diabetes mellitus, statistically significant *p*-values were printed in bold.

	EPP	LPS	Palliative	*p*-Value
Any comorbidities	5 (55.6%)	24 (60.0%)	40 (63.4%)	0.15
Arterial hypertension	4 (44.4%)	17 (42.5%)	25 (39.7%)	**0.034**
Coronary artery disease	1 (11.1%)	6 (15.0%)	12 (19.0%)	0.70
History of cancer	1 (11.1%)	4 (10.0%)	11 (17.4%)	0.43
NIDDM	0 (0.0%)	4 (10.0%)	7 (11.1%)	0.61
Cardiomyopathy	0 (0.0%)	2 (5.0%)	6 (9.5%)	0.25
Atrial fibrillation	1 (11.1%)	3 (7.5%)	4 (6.3%)	0.84
Hyperthyroidism	0 (0.0%)	1 (2.5%)	4 (6.3%)	0.23
History of stroke	1 (11.1%)	0 (0.0%)	1 (1.6%)	0.64
Pancreatitis	0 (0.0%)	1 (2.5%)	0 (0.0%)	0.51
Hodgkin’s disease	0 (0.0%)	0 (0.0%)	1 (1.6%)	0.57
Unknown comorbidities	0 (0.0%)	0 (0.0%)	1 (1.6%)	**0.038**

**Table 3 cancers-14-02245-t003:** Postoperative complications in the whole cohort. The *p*-values and hazard ratios were determined by univariate Cox regression. ARDS acute respiratory distress syndrome, statistically significant *p*-values were printed in bold.

Complications	EPP	LPS	*p*-Value
Any complications	8 (88.9%)	21 (52.5%)	**0.02**
30-day mortality	0 (0.0%)	0 (0.0%)	
Blood transfusion required	1 (11.1%)	10 (25.0%)	**0.033**
Postoperative air leak	0 (0.0%)	8 (20.0%)	0.59
Atrial fibrillation	5 (55.6%)	3 (7.5%)	0.30
Empyema	4 (44.4%)	0 (0.0%)	0.49
Pneumonia	1 (11.1%)	2 (5.4%)	0.64
Pulmonary edema	0 (0.0%)	1 (2.5%)	**0.014**
Cardiac decompensation	1 (11.1%)	1 (2.5%)	**<0.001**
Postoperative hemorrhage	1 (11.1%)	1 (2.5%)	0.58
Wound healing disorder	2 (22.2%)	0 (0.0%)	0.22
Postoperative abscess	0 (0.0%)	1 (2.5%)	**0.014**
Wound infection	0 (0.0%)	1 (2.5%)	0.81
Hypertensive derailment	0 (0.0%)	1 (2.5%)	0.36
Hyperglycemia	0 (0.0%)	1 (2.5%)	0.36
Urinary tract infection	0 (0.0%)	1 (2.5%)	0.14
Renal insufficiency	1 (11.1%)	0 (0.0%)	0.12
ARDS	0 (0.0%)	1 (2.5%)	0.41
Right-sided acute cardiac decompensation	0 (0.0%)	1 (2.5%)	0.37
Hyperfibrinolysis	0 (0.0%)	1 (2.5%)	0.37
Rupture of the diaphragmatic patch	1 (11.1%)	0 (0.0%)	0.89
Gastric herniation	1 (11.1%)	0 (0.0%)	0.89

**Table 4 cancers-14-02245-t004:** Survival rates in months for different subgroups and interventions. In total, 112 patients were analyzed. EPD extended pleurectomy/decortication, EPP extrapleural pneumonectomy, LPS lung-preserving surgery.

	Median OS (95% CI) (Months)	Range (Months)	N
All patients	16.9 (13.4; 20.4)	0.2–184.1	112
Sex			
Female	30.0 (21.9; 38.1)	1.2–184.1	24
Male	14.8 (11.2; 18.4)	0.2–100.9	88
Pathological Stage			
Stage I	42.3 (17.8; 66.8)	10.6–89.4	15
Stage II	18.1 (16.0; 20.2)	1.0–92.5	16
Stage III	17.4 (15.1; 19.7)	1.3–100.9	31
Stage IV	8.5 (5.0; 12.0)	0.2–62.8	44
Unknown	22.2 (4.9; 39.5)	1.2–184.1	6
Histological subtype			
Epitheloid	20.3 (16.9; 23.7)	0.5–184.1	87
Non-epitheloid	6.8 (3.2; 10.4)	0.2–27.7	25
Surgical treatment			
Multimodality therapy	22.7 (17.2; 18.1)	5.1–184.1	57
Trimodality therapy	30.8 (7.2; 54.3)	11.9–89.4	10
Neoadjuvant chemotherapy + surgery	21.4 (14.1; 28.7)	5.1–100.9	39
Surgery + adjuvant chemotherapy	17.8 (16.3; 19.3)	10.6–184.1	8
Surgical intervention			
EPP	12.3 (11.1; 13.5)	6.0–30.8	9
EPD or P/D (LPS)	25.5 (37.3; 81.6)	7.0–184.1	40
Non-surgical treatment			
Chemotherapy +/− radiotherapy	11.3 (7.9; 14.7)	0.8–63.3	36
Other	5.4 (1.5; 9.3)	0.2–62.8	19

**Table 5 cancers-14-02245-t005:** Analysis of the factors of possible prognostic significance in our cohort, as determined by univariate Cox regression. CD cardiac decompensation, CI confidence interval, CHT chemotherapy, LPS lung-preserving surgery, OS overall survival, PE pulmonary edema, RCC red cell concentrate; statistically significant *p*-values were printed in bold.

Univariate Analysis							
Factor		Median OS (Months)	1-Year OS % (*n*)	3-Year OS % (*n*)	5-Year OS % (*n*)	*p*-Value	Hazard Ratio (95% CI)
Age (per 1-year increase)						**0.011**	
Age < 55 years	Younger than 55 (22) Older than (55)	25.4 13.3	90.9 (20) 58.5 (48)	40.9 (8) 14.3 (11)	24.5 (2) 14.3 (11)	**0.007**	0.47 (0.27–0.81)
Epithelioid vs. non-epithelioid subtype	Epithelioid Other	20.3 6.8	73.7 (60) 32.0 (8)	25.9 (19) 0	17.1 (10) 0	**<0.001**	0.25 (0.15–0.41)
Sex (female vs. male)	Male (88) Female (24)	14.8 30.0	60.3 (50) 78.4 (16)	12.5 (10) 41.5 (7)	9.9 (5) 25.9 (3)	**0.021**	0.54 (0.32–0.91)
Tumor side (right vs. left)	Right (44) Left (68)	18.1 16.1	66.7 (28) 62.4 (40)	23.8 (10) 16.9 (9)	12.9 (4) 10.9 (5)	0.81	0.95 (0.63–1.43)
Tumor stage (I–IV)	I (15) II (16) III (31) IV (44)	42.3 18.1 17.4 8.5	93.3 (14) 73.3 (11) 76.8 (23) 39.8 (17)	56.2 (7) 13.3 (2) 23.4 (7) 5.9 (2)	24.1 (2) 6.7 (1) 20.0 (4) 5.9 (1)	**0.002**	1.09 (0.98–1.20)
Surgical intervention vs. non-surgical intervention	Surgical (55) Non-surgical (57)	21.4 11.1	84.9 (44) 43.7 (23)	28.7 (13) 11.4 (6)	21.8 (9) 5.1 (2)	**<0.001**	0.43 (0.28–0.65)
Radicality of resection (R1 vs. R2)	R1 (45) R2 (10)	23.3 16.9	84.4 (38) 87.5 (7)	31.5 (12) 12.5 (1)	26.0 (8) 0	0.063	2.11 (0.96–4.66)
LPS vs. other interventions	LPS (46) Other (66)	21.4 11.4	90.90 (40) 45.3 (28)	34.8 (13) 34.9 (6)	26.5 (9) 4.3 (2)	**<0.001**	0.62 (0.50–0.77)
Trimodal therapy vs. other interventions	Trimodal (10) Other (102)	30.8 15.8	90.0 (9) 61.5 (59)	50.0 (5) 16.8 (15)	37.5 (2) 10.7 (8)	0.076	0.52 (0.25–1.07)
Multimodal therapy (adjuvant or neoadjuvant chemotherapy + surgery with or without radiotherapy) vs. other interventions	Multimodal (57) Other (55)	22.7 9.9	85.5 (47) 41.6 (22)	29.3 (14) 9.9 (5)	22.8 (10) 4.0 (2)	**<0.001**	0.37 (0.25–0.56)
Surgical intervention + chemotherapy vs. other intervention	Surgery and CHT (8) Other (104)	17.8 15.8	75.0 (6) 63.3 (62)	25.0 (2) 19.5 (17)	12.5 (1) 13.2 (9)	0.53	0.78 (0.36–1,69)
Arterial hypertension	Hypertension (46) No hypertension (66)	16.1 17.8	62.5 (28) 65.3 (40)	34.9 (3) 29.9 (16)	4.5 (2) 20.1 (10)	**0.034**	1.56 (1.03–2.35)
Occurrence of any complications	Complications (31) No complications (24)	17.8 32.7	90.9 (20) 80.6 (25)	47.3 (9) 16.1 (5)	36.0 (6) 12.1 (2)	**0.020**	1.22 (0.81–1.84)
RCC application required	RCC required (11) No RCC required (44)	20.0 25.4	81.8 (9) 85.8 (36)	9.1 (1) 33.9 (13)	028.0 (9)	**0.033**	1.30 (0.71–2.41)
Cardiac decompensation	CD (2) No CD (53)	6.0 22.7	0 88.2 (45)	0 33.3 (13)	0 22.7 (9)	**<0.001**	6.45 (2.23–18.65)
Pulmonary edema	PE (1) No PE (54)	10.6 21.4	0 86.6 (45)	0 29.2 (13)	0 22.3 (9)	**0.014**	1.04 (0.26–4.24)
Postoperative abscess	Abscess (1) No abscess (54)	10.6 21.4	0 86.6 (45)	0 29.2 (13)	0 22.3 (9)	**0.014**	3.08 (0.42–22.59)

**Table 6 cancers-14-02245-t006:** Factors of possible prognostic significance in surgically treated patients. Statistically significant *p*-values were printed in bold.

Factor	*p*-Value
Age	0.39
Epithelioid vs. non-epithelioid subtype	**0.008**
EPD or P/D vs. EPP	**0.017**
Sex (female/male)	**0.032**
Lymph node state (N2 neg./pos.)	0.95
Radicality of resection (R1/R2)	0.63
Trimodal therapy vs. other interventions	0.39
Tumor side (right/left)	0.58
Tumor stage (I–IV)	0.29

**Table 7 cancers-14-02245-t007:** Multivariate analysis of factors that had a significant impact on patients’ survival in the univariate analysis. The *p*-values and hazard ratios were determined by multivariate Cox regression. LPS lung-preserving surgery, RCC red cell concentrate; statistically significant *p*-values were printed in bold.

Multivariate Analysis			
Factor	Hazard Ratio	95% CI	*p*-Value
Age (per 1-year increase)	0.98	0.96–1.02	0.44
Age < 55 years vs. Age > 55 years	0.64	0.34–1.96	0.64
Epithelioid vs. non-epithelioid subtype	0.35	0.20–0.62	**<0.001**
Sex (female vs. male)	0.75	0.4–1.41	0.371
Tumor stage (I–IV)	1.29	1.05–1.60	**0.014**
Surgical intervention vs. non-surgical intervention	0.00014	<0.0001–>1000	0.87
LPS vs. other interventions	0.81	0.53–1.22	0.31
Multimodal therapy (adjuvant or neoadjuvant chemotherapy + surgery with or without radiotherapy) vs. other interventions	<0.0001	<0.0001–>1000	0.86
Arterial hypertension	1.75	1.08–2.86	**0.025**
Occurrence of any complications	1.31	0.75–2.26	0.34
RCC application required	1.50	0.67–3.36	0.33
Cardiac decompensation	5.43	1.41–20.88	**0.014**
Pulmonary edema	2104.0	<0.0001–>1000	0.89
Postoperative abscess	0.00045	<0.0001–>1000	0.89

**Table 8 cancers-14-02245-t008:** Overview of median survival of MPM in the literature.

Author	Year	n	Stage I	Stage II	Stage III	Stage IV
			Months	Months	Months	Months
Sugarbaker	1998	120	22	17	11	
Rush	1999	231	30	19	10	8
Flores	2008	663	38	19	11	7
Buduhan	2009	49	17	33	31	24
Rena	2012	77	28	18		
Bölükbas	2012	78			21	8
IASLC Database (Rusch)	2012	3101	21	19	16	12
Sezer	2013	54		11	19	11

## Data Availability

The data presented in this study are available on request from the corresponding author. The data are not publicly available due to the valid European General Data Protection Regulations.

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
