# Peer review of "Multimodal Treatment of Malignant Pleural Mesothelioma: Real-World Experience with 112 Patients"

_cancers, 2022, doi:10.3390/cancers14092245_

Round 1

Reviewer 1 Report

The Authors have retrospectively analysed a cohort of 112 patients with mesothelioma that underwent surgery at a high-volume center. The paper is well written and easy to read. The results of the analysis are consistent with the literature. In statistical  analysis, combined chemotherapy and surgery, epithelioid tumor subtype, early tumor stage and the absence of relevant comorbidities were found to be favorable factors for survival. The Authors conclude that best possible treatment for malignant pleural mesothelioma should incorporate multiple therapeutic approaches.

I suggest to add some clarifications on some points, as follows:

-The Authors should better clarify the criteria used to guide the clinical decision of the workflow (i.e.: induction chemotherapy vs adjuvant chemo? Why no patients after adjuvant chemo were referred for radiotherapy? After induction chemo on which basis the Clinicians decided to perform surgery or adjuvant radiotherapy and why is it called adjuvant radiotherapy. For this subset the Authors state that adjuvant radiotherapy where applicable, but all the patients that did not perform surgery were treated with adjuvant radiotherapy. Finally the Authors should argument why only a very small percentage of the patients were treated with trimodal therapy (8.9%).

-The use of Radiotherapy in this setting is extremely complex and require a significant experience of the Clinicians. The Authors should better describe the technical parts of radiotherapy (volume, doses, differences between patients treated with surgery (and for this subset the differences between P/D and EPP) and patients treated without  surgery (maybe in a palliative setting?). The Authors should also clarify whether the patients treated with radiotherapy were treated in the same center of were referred to different centers, and the experience of the radiotherapy centers for this disease.

-The Authors should add a table of the side effects of the whole cohort of patients, divided according to the treatments used.

Author Response

Dear Reviewer 1,

thank you very much for your very accurate and highly insightful review. Please find our answers below.

“The Authors should better clarify the criteria used to guide the clinical decision of the workflow (i.e.: induction chemotherapy vs adjuvant chemo? Why no patients after adjuvant chemo were referred for radiotherapy? After induction chemo on which basis the Clinicians decided to perform surgery or adjuvant radiotherapy and why is it called adjuvant radiotherapy. For this subset the Authors state that adjuvant radiotherapy where applicable, but all the patients that did not perform surgery were treated with adjuvant radiotherapy.”

In general, all patients who were eligible (and consented for it) received induction chemotherapy. In case of the eight patients who were treated with adjuvant chemotherapy after surgery, the diagnosis of MPM was not anticipated based on clinical and radiological assessment preoperatively [see second paragraph in Results section]. For clarification, we added a sentence to the methods section. Of those eight patients, none has received radiotherapy (based on treating physicians’ discretion and usually because of poor performance-status of the patient).

As you correctly pointed out, Figure 2 is unfortunately misleading. Not all patients who did not receive surgery were treated with radiotherapy. As the focus of the paper was surgical, the follow-up of non-surgically-treated patients was initially more minimalistic – so the term of „where applicable“ or „if feasible“ describing only the possibility that radiotherapy was administered. We thus sifted again through our database for those 36 patients: Only 10 of those patients’ received radiotherapy after inductive chemotherapy (see also below at #2). We now created a modified version of Figure 2 and changed that part in the results section.

“Finally the Authors should argument why only a very small percentage of the patients were treated with trimodal therapy (8.9%).”

Only a very small percentage of the patients were treated with trimodal therapy, because only a minority of patients had a sufficient performance status to sustain a third treatment modality immediately after bimodal therapy (induction chemotherapy plus surgery), namely radiotherapy, which is in our experience very demanding for the patient.

“The use of Radiotherapy in this setting is extremely complex and require a significant experience of the Clinicians. The Authors should better describe the technical parts of radiotherapy (volume, doses, differences between patients treated with surgery (and for this subset the differences between P/D and EPP) and patients treated without surgery (maybe in a palliative setting?). The Authors should also clarify whether the patients treated with radiotherapy were treated in the same center of were referred to different centers, and the experience of the radiotherapy centers for this disease.”

Our patients received radiotherapy at an academic referral-center (University Hospital Vienna) – we added some information to the methods section.

In Trimodal Setting all patients received intensity modulated radiotherapy with a dose of 54 to 60 Gy (single doses of 2 Gy). We added this information to the methods section. In the 10 patients who were not treated with surgery but have received radiotherapy after induction therapy the information on exact dose is lacking: 4 patients received a dose of 30-33Gy, 1 received 54Gy for the others. Unfortunately, no information on the exact radiation protocol is available retrospectively.

“The Authors should add a table of the side effects of the whole cohort of patients, divided according to the treatments used.”

A table of side effects for the whole cohort of patients, divided by treatment, has been added.  

Reviewer 2 Report

The work by Arnulf Holzknecht, et al. provided a retrospective analysis of a large and precious clinical cohort of malignant pleural mesothelioma (MPM), a highly aggressive malignant disease that is rare with a lack of effective treatment. Generally, the work was well-structured and well-conceived, but some minor issues need to be addressed.

  1. The authors concluded that multimodality therapy including surgery significantly prolonged OS of MPM patients as compared to multimodal therapy without surgery. This should be interpreted with caution, as the multivariate Cox analysis that adjusted for confounders did not identify the addition of surgery as an independent factor influencing the OS. Although the univariate survival analysis did show the addition of surgery was associated with improved OS, the surgery group had a significantly higher percentage of the epithelioid subtype, which is a well-known marker predictive of better survival in MPM patients.
  2. The combination of other therapies with immunotherapies is increasingly appreciated in MPM (PMID: 33240401). Could the authors also add some relevant discussion?

Author Response

Dear Reviewer 2,

thank you very much for your very accurate and highly insightful review. Please find our answers below.

“The authors concluded that multimodality therapy including surgery significantly prolonged OS of MPM patients as compared to multimodal therapy without surgery. This should be interpreted with caution, as the multivariate Cox analysis that adjusted for confounders did not identify the addition of surgery as an independent factor influencing the OS. Although the univariate survival analysis did show the addition of surgery was associated with improved OS, the surgery group had a significantly higher percentage of the epithelioid subtype, which is a well-known marker predictive of better survival in MPM patients.”

Thank you, this is a very important comment. To highlight this important limitation of our analysis, we amended the Discussion accordingly, and pointed out the results of the univariate analysis again.

“The combination of other therapies with immunotherapies is increasingly appreciated in MPM (PMID: 33240401). Could the authors also add some relevant discussion?”

How immunotherapy can be implemented within a multimodality treatment algorithm in MPM and possibly improve OS, is an important and thrilling question indeed. Unfortunately, we have not analyzed systemic therapy in case of disease recurrence in our study (so only neoadjuvant and adjuvant chemotherapy was analyzed). For the manuscript we now added a sentence to the Discussion to point out that limitation. Moreover, to highlight the importance and the growing interest in implementing checkpoint-inhibitors into MPM treatment protocols, we cited the Yang et al. paper you suggested, and Checkmate-743. (In fact, we are planning to have a look into the treatment regimens of disease recurrence and the respective outcome in our long-term survivors in the future.)

Reviewer 3 Report

Thank you for the opportunity to review this manuscript.
It is a single-centre retrospective cohort study of prognostic factors in a series of mesotheliomas.
The work is interesting, but suffers from several points that need to be rectified: 
- The statistical analysis of correlation was only done using the X2 test. The statistical analysis of the correlation was done only with the X2 test, but under certain conditions it is not acceptable to use this test and it should be replaced by a Fisher exact test.
- Not all the statistical tests used are described in this report.
- It would be interesting to reclassify tumours according to the latest version of the TNM, 8th edition.
- Other histopathological subtype: the authors can specify this type as there is only one case
- Table 1: N2 status: part of the table is missing
- Table 2 is not readable and is a combination of two tables. Furthermore, the statistical tests used are not specified, there is a p-value that comes from nowhere.
- In what seems to be table 4, the authors could specify the p-value of the log-rank test, which would be interesting
- The statistics for univariate and multivariate analysis are not clear. Hazard ratios are not specified for the univariate analysis. The method of selecting the variables included for the multivariate analysis is unclear.
- No notion of histological grading is specified, although this is a major prognostic factor among epithelioid mesotheliomas. The authors could discuss this and cite articles dealing with this prognostic notion.
All in all, the idea is interesting, but the analysis of the results is lacking in quality, especially concerning the statistical analysis. 
I did not detect any plagiarism or image falsification.

Author Response

Dear Reviewer 3,

thank you very much for your very accurate and highly insightful review. Please find our answers below.

“The statistical analysis of correlation was only done using the X2 test. The statistical analysis of the correlation was done only with the X2 test, but under certain conditions it is not acceptable to use this test and it should be replaced by a Fisher exact test.”

Thank you for your very valuable suggestion. All correlations have now been done with Fisher’s exact test, and results and discussion have been modified accordingly.

“Not all the statistical tests used are described in this report.”

In order to describe all used statistical tests completely, the methods section has now been amended.

“It would be interesting to reclassify tumours according to the latest version of the TNM, 8th edition.”

We agree with you that this would be interesting, and therefore we performed a sampling comparison between OS results based on the 7th edition and on the 8th edition, but since the results were completely similar (data not shown), we saw no point in reorganizing the whole analysis. However, the most striking difference between 7th and 8th TNM edition is the re-definition of N factors, with a disappearance of former N3 and subsequent accommodation of ‘old’ N3 into ‘new’ N2, and furthermore, a combining of all ipsilateral affected lymph nodes under ‘new’ N1. Thus, in order to avoid possible misunderstandings regarding our ‘old’ N2 data (we never had patients with ‘old’ N3), we removed all correlations with the ‘old’ N2 from our analysis.

“Other histopathological subtype: the authors can specify this type as there is only one case.”

The other subtype you asked for was a lymphohistiocytoid mesothelioma. It has been added to the Tables.

“Table 1: N2 status: part of the table is missing”; Table 2 is not readable and is a combination of two tables. Furthermore, the statistical tests used are not specified, there is a p-value that comes from nowhere.”

We apologize: due to formatting problems with the available template, the tables had to be submitted as pictures (the complete tables had then been submitted separately), and not only this, but they had also been partially truncated. This problem is now solved, and the improved complete tables containing those statistical tests, that are described in the methods section, have been added to the manuscript. Furthermore, the combined table had been separated.

“In what seems to be table 4, the authors could specify the p-value of the log-rank test, which would be interesting.”

All p values of interest are now specified in the completely renewed tables.

“The statistics for univariate and multivariate analysis are not clear. Hazard ratios are not specified for the univariate analysis. The method of selecting the variables included for the multivariate analysis is unclear.”

The adopted statistical tests are now fully described, including the presentation of Hazard ratios for the univariate analysis and the description of the method of selecting the variables for the multivariate analysis.

“No notion of histological grading is specified, although this is a major prognostic factor among epithelioid mesotheliomas. The authors could discuss this and cite articles dealing with this prognostic notion.”

We completely agree with you, that the correlation of histological grading with OS would be a very important enhancement of our results. Unfortunately, in the past the pathologists in our center did never routinely determine histological grading, and still not do. Re-assess all histological specimen would theoretically be possible, but not feasible both in terms of time and economic resources. We can do better in the future, and as a matter of fact, our pathologists now promised to prospectively assess the grading of all incoming samples.

As for the present paper, we will acknowledge this limitation by adding a respective sentence to the discussion.

Round 2

Reviewer 1 Report

The Authors have responded to previous comments. The manuscript is now suitable for publication.

Reviewer 3 Report

The authors have responded to all my comments.
I have no further comments.